# Common Pitfalls in the Management of Patients with Micronutrient Deficiency: Keep in Mind the Stomach

**DOI:** 10.3390/nu13010208

**Published:** 2021-01-13

**Authors:** Marilia Carabotti, Bruno Annibale, Edith Lahner

**Affiliations:** Department of Medical-Surgical Sciences and Translational Medicine, Sant’Andrea Hospital, University Sapienza, 00189 Rome, Italy; marilia.carabotti@uniroma1.it (M.C.); bruno.annibale@uniroma1.it (B.A.)

**Keywords:** anemia, atrophic gastritis, ascorbic acid, autoimmune gastritis, bariatric surgery, calcium deficiency, cobalamin deficiency, *Helicobacter pylori*, iron deficiency, magnesium deficiency, micronutrients, pernicious anemia, proton pump inhibitors

## Abstract

Micronutrient deficiencies are relatively common, in particular iron and cobalamin deficiency, and may potentially lead to life-threatening clinical consequences when not promptly recognized and treated, especially in elderly patients. The stomach plays an important role in the homeostasis of some important hematopoietic micronutrients like iron and cobalamin, and probably in others equally important such as ascorbic acid, calcium, and magnesium. A key role is played by the corpus oxyntic mucosa composed of parietal cells whose main function is gastric acid secretion and intrinsic factor production. Gastric acid secretion is necessary for the digestion and absorption of cobalamin and the absorption of iron, calcium, and probably magnesium, and is also essential for the absorption, secretion, and activation of ascorbic acid. Several pathological conditions such as *Helicobacter pylori*-related gastritis, corpus atrophic gastritis, as well as antisecretory drugs, and gastric surgery may interfere with the normal functioning of gastric oxyntic mucosa and micronutrients homeostasis. Investigation of the stomach by gastroscopy plus biopsies should always be considered in the management of patients with micronutrient deficiencies. The current review focuses on the physiological and pathophysiological aspects of gastric acid secretion and the role of the stomach in iron, cobalamin, calcium, and magnesium deficiency and ascorbate homeostasis.

## 1. Introduction

Micronutrient deficiencies are relatively common worldwide and are recognized as a global public health issue [1,2]. In particular, deficiencies of the erythropoietic micronutrients, iron, and cobalamin may potentially lead to life-threatening clinical consequences when not promptly recognized and treated, especially in elderly patients [3]. The stomach plays an important role in the homeostasis of several important hematopoietic micronutrients such as cobalamin and iron, and probably in others equally important like ascorbic acid, calcium, and magnesium [4,5,6].

A key role is played by the gastric oxyntic mucosa composed of parietal cells whose main function is gastric acid secretion and production of intrinsic factor. Gastric acid secretion is necessary for the digestion and absorption of cobalamin and the absorption of iron, calcium, and probably magnesium, and is also essential for the secretion, activation, and absorption of ascorbic acid.

Cobalamin absorption from food requires the presence of gastric acid for at least two steps of this complex process: to release into the stomach the strictly protein-bound dietary cobalamin and to enhance the affinity of unbound cobalamin for salivary R proteins instead of intrinsic factor; besides, for the ileal absorption of cobalamin, intrinsic factor is necessary, which is also secreted by gastric parietal cells [5,7,8]. For the absorption of the non-haemic ferric iron, the most frequent form of iron in the Western diet, gastric acid is necessary to maintain solubilization, to avoid precipitation, and to make possible reduction and chelation by ascorbic acid, an important promoter of iron absorption [9]. Gastric acidity seems to play a role also in the homeostasis of ascorbic acid, the reduced form of vitamin C, as in a non-acidic environment, it is converted to dehydroascorbic acid, a less active form [5,10]. Ascorbic acid further promotes iron absorption, thus decreased bioavailability of this micronutrient may imply negative effects on iron absorption [11].

Several pathological conditions such as *Helicobacter pylori*-related gastritis and corpus atrophic gastritis [5,6,12], as well as antisecretory drugs, especially proton pump inhibitors [13], and also gastric surgery [14,15,16], may interfere with the normal functioning of the gastric oxyntic mucosa and micronutrients homeostasis. The hypochlorhydric stomach in patients with autoimmune and non-autoimmune corpus atrophic gastritis has been shown to allow the survival and colonization of bacterial strains other than *Helicobacter pylori*, and it is suggested that gastric dysbiosis is related to the increased risk of gastric cancer in this condition [17]. The specific role of gastric dysbiosis in the impairment of micronutrient bioavailability or absorption has not been reported so far.

This often forgotten link between the stomach and micronutrients may have a relevant impact on a huge amount of people. Taking into consideration the high prevalence of *Helicobacter pylori* infection all over the world, its role in micronutrient malnutrition may have paramount clinical implications: albeit a decreasing trend has been shown globally in the last years, yet more than 50% of people in the world are infected by *Helicobacter pylori*¸ and in Africa, Central and South America, a large part of the population is infected by *Helicobacter pylori* [18]. This is even more true when taking into account that in poorly industrialized, developing regions, a high prevalence of *Helicobacter pylori* infection and micronutrient malnutrition are concomitant [19,20]. The estimated prevalence of corpus atrophic gastritis ranges up to 24% based on serology and 34% based on histology [12]. Moreover, antisecretory agents, in particular, proton pump inhibitors are one of the most widely used drugs in the Western world [21], and are the mainstay of treatment of gastroesophageal reflux disease and its complications, including short- and long-term therapy of *Helicobacter pylori*-negative peptic ulcers; healing and prevention of drug-associated gastric ulcers; co-therapy with endoscopic procedures to control upper digestive bleeding; medical treatment of Zollinger Ellison syndrome, and; eradication of *Helicobacter pylori* infection together with two or more antibiotics [22]. In the last years, gastric surgical procedures are increasingly performed to treat obesity. According to a 2013 worldwide survey, the total number of bariatric procedures performed worldwide in 2013 was 468,609, with the vast majority (95.7%) carried out laparoscopically, and the highest number (*n* = 154,276) performed in the USA/Canada. Worldwide, the most common procedure was Roux-en-Y gastric bypass (45%) followed by sleeve gastrectomy (37%) [23].

During the diagnostic work-up of patients with micronutrient deficiency, the investigation of the stomach by gastroscopy with biopsies should always be considered due to its potential causal role. The current review focuses on the physiological aspects of gastric acid secretion and the role of the stomach in iron, cobalamin, calcium, and magnesium deficiency and ascorbate homeostasis.

## 2. Physiology of Gastric Oxyntic Mucosa: Secretion of Gastric Acid and Intrinsic Factor

The human stomach has both exocrine and endocrine secretory activities. The proximal part of the stomach (fundus and corpus) is characterized by oxyntic glands (composed of: mucous cell [mucus]; parietal cell [hydrochloric acid and intrinsic factor]; D cell [somatostatin]; chief cell [pepsinogen and leptin]; enterochromaffin-like cell [histamine]; A-like or Gr Cell [ghrelin]), and the distal part (antrum) presented pyloric glands (composed of: mucous cell [mucus]; G cell [gastrin]; D cell [somatostatin]; enterochromaffin cell [atrial natriuretic peptide]) [24,25]. Among these substances, hydrochloric acid (HCl) and intrinsic factor (IF) is essential for the proper absorption of some micronutrients.

HCl is produced by proton pumps, membrane proteins with ATPase activity (H+, K + ATPase) located on the parietal cells. The secretion of HCl is finely regulated by the coordination of several signals (endocrine, paracrine, and neurocrine) acting directly and/or indirectly on its release [26,27]. Table 1 provides a summary of the principal stimulants and inhibitors of HCl secretion and their relative mechanisms of action [25,26,27,28].

Similarly, IF secretion is stimulated via all pathways known to stimulate gastric acid secretion: histamine, gastrin, and acetylcholine [29].

Thus, an inflammatory injury eventually leading to apoptosis of gastric glands or pharmacological inhibition of gastric stimulants such as H_2_ blockers, or of specific functions like proton pump inhibitors, give arise to impairment of the functioning of these highly specialized cells. When gastric parietal cells are involved in this process, this may give rise to reduced production of gastric acid and/or intrinsic factor. Gastric secretion may be reduced also by another mechanism: when important stimulants of gastric acid secretion are lacking, for example, antral G cells after partial gastric resection, gastric acid secretion is heavily impaired and intragastric pH will increase.

## 3. The Role of the Stomach in Non-Bleeding-Related Iron Deficiency

Dietary iron is available in two different forms, such as heme and non-heme iron [11]. The heme iron (ferrous form) contained in the hemoglobin and myoglobin of meat, includes 5–10% of the total dietary iron in the Western diet. It is easily absorbed by the small intestine mucosal cells through a surface receptor. Non-heme iron (ferric iron) contained in cereals, vegetables, legumes, and fruits represents 80% of the dietary iron [30]. Non-heme iron is transported across the apical membrane of the enterocyte by divalent metal-ion transporter 1 (DMT1) and is exported into the circulation via ferroportin 1 (FPN1) [31]. This latter form is less-absorbable, not soluble, and precipitates at pH 3, being absorbable only when reduced to the ferrous or chelated form [11]. In physiological conditions, HCl and ascorbic acid play a crucial role in iron absorption promoting the reduction of non-heme iron from its ferric form to the ferrous form. Moreover, at acid pH (<3), ascorbic acid constitutes soluble chelates with ferric iron-reducing its polymerization and precipitation [9]. Keeping in mind these mechanisms, it is understandable how some pathological gastric conditions might affect iron absorption leading to iron deficiency and eventually iron deficiency anemia.

*Helicobacter pylori* infection is the main etiologic factor of chronic gastritis [32,33,34]. This infection is one of the most common in humans worldwide and it is estimated that more than half of the world population is infected [35]. It is highly prevalent in developed countries and in Europe, it has been estimated that a variable proportion of subjects ranging from 11% to 84.2% are infected [36]. Sero-epidemiological studies published several years ago supported the association between *Helicobacter pylori* infection and low ferritin values [5,37,38,39,40,41,42].

More recently, other authors worldwide reported the association between *Helicobacter pylori* infection and iron deficiency anemia. A recent study conducted in Mexico found a 7.8% prevalence of *Helicobacter pylori* infection in iron deficiency anemia patients referred to an academic hematology center [43], while an Egyptian study showed a higher prevalence of *Helicobacter pylori* infection in refractory iron deficiency anemia (61.5%) and a significant correlation between administration of anti-*Helicobacter pylori* treatment plus iron and improvement in terms of hemoglobin, mean corpuscular volume, iron, and ferritin levels [44]. A recent systematic review and meta-analysis showed that compared to uninfected subjects, *Helicobacter pylori*-infected individuals showed an increased likelihood of iron deficiency anemia (OR 1.72; 95% CI 1.23–2.42); and iron deficiency (OR 1.33; 95% CI 1.15–1.54), reporting increased ferritin levels following anti-*Helicobacter pylori* eradication therapy plus iron therapy as compared with iron therapy alone [45].

Among the hypothesized mechanisms determining the association between *Helicobacter pylori* and iron deficiency and/or iron deficiency anemia (i.e., virulence factors, competition with the host for the acquisition of alimentary iron), the most accredited hypothesis is that particular gastritis patterns with a more extensive degree of mucosal inflammation (pangastritis or corpus predominant gastritis) compared to a more limited, antrum-restricted gastritis may lead to iron malabsorption [11]. The corpus gastric inflammation causes an increase in intragastric pH, which was significantly higher in anemic patients than in controls (median value of 5.7 vs. 2) [46]. Thus, it is plausible that the gastric corpus involvement leads to reduced availability of alimentary iron due to decreased acidity.

Similarly, corpus atrophic gastritis, a condition characterized by atrophy of oxyntic glands due to a long-standing *Helicobacter pylori* infection or autoimmunity, determines reduced gastric acid production with consequent iron malabsorption and eventually iron deficiency anemia [12]. Even if corpus atrophic gastritis has been traditionally considered as synonymous with pernicious anemia, and thus of cobalamin (vitamin B_12_) malabsorption, already in the 1960s the correction of iron absorption by adding gastric juice in patients with pernicious anemia was observed [47,48], suggesting the role of HCl in the absorption process. Firstly, Dickey et al. showed that about 20% of iron deficiency anemia patients had corpus atrophic gastritis, diagnosed by both gastric histology and high serum levels of gastrin [49]. Later, other authors reported that 19.5% to 26% of iron deficiency anemia patients without gastrointestinal symptoms have corpus atrophic gastritis [50,51]. Similar results were obtained by Hershko et al., who found that 27% of iron deficiency anemia patients in absence of evident gastrointestinal diseases, presented autoimmune atrophic gastritis [52]. In a more recent observational single-center study by Zilli et al., iron deficiency and iron-deficiency anemia were present in 34% and 13.1% of atrophic autoimmune gastritis patients respectively [6].

Thus, in patients with iron deficiency or iron-deficiency anemia without manifest or occult bleeding, an accurate diagnosis of *Helicobacter pylori* infection and related gastritis is mandatory and the presence of corpus atrophic gastritis needs to be ruled out [35,53]. Therefore, gastroscopy with standard bioptic sampling is necessary. Superficial *Helicobacter-pylori* gastritis requires eradication treatment based on antibiotics and cure of infection generally leads to regression of iron deficiency or iron-deficiency anemia. Once corpus atrophic gastritis is present, the cure of *Helicobacter pylori* infection, when present, uncommonly gives rise to regression of anemia, and oral or intravenous iron supplementation is necessary depending on the severity of anemia.

## 4. The Role of the Stomach in Cobalamin Deficiency

The main food source of vitamin B_12_ (cobalamin) is animal products. For vitamin B_12_ absorption, intact gastric corpus mucosa that harbors oxyntic glands including parietal cells is necessary. Vitamin B_12_ absorption is complex with several different steps requiring gastric acid and a transporter protein, called intrinsic factor, both produced by gastric oxyntic parietal cells [54]. After its intake, vitamin B_12_ is firstly released from the food carrier protein by proteolysis at acidic pH in the stomach and then bound to haptocorrin, a salivary protein protecting the vitamin from acid degradation. In the duodenum, haptocorrin is degraded and vitamin B_12_ binds to gastric intrinsic factor. This vitamin B_12_-intrinsic factor complex is then uptaken by the enterocytes of the distal ileum mediated by the cubam receptor via receptor-mediated endocytosis [54,55]. Vitamin B_12_ then exits via the basolateral membrane of enterocytes and binds to its blood carrier transcobalamin delivering the vitamin to the cells. The largest amount of vitamin B_12_ is stored in the liver, a small amount is excreted in the bile and takes part in the enterohepatic circulation [54,55,56].

There are several pathological conditions associated with reduced gastric acid and/or intrinsic factor secretion, some of them are potentially reversible when treatment is timely, others are mostly irreversible due to permanent damage of the oxyntic mucosa.

Chronic *Helicobacter pylori*-related gastritis involving the corpus mucosa may lead to functional inhibition of gastric parietal cells due to mucosal inflammation and its products, leading to vitamin B_12_ maldigestion, also called “food cobalamin malabsorption”, as the vitamin cannot be released from the food carrier protein due to the lack of pepsin activation in a non-acidic stomach [7,57,58,59,60]. After the cure of *Helicobacter pylori* infection, parietal cell function may be restored and vitamin B_12_ absorption is normalized.

A systematic review investigated the possible link between *Helicobacter pylori* infection and reduced micronutrients beyond iron: pooled standardized mean differences (SMD) in micronutrient levels showed a positive association with *Helicobacter pylori* infection for serum cobalamin levels (SMD 0.744), and a positive effect of eradication treatment, which lead to the increase of serum cobalamin levels (SMD 1.910) [4], thus supporting the idea that impaired gastric acid secretion caused by *Helicobacter pylori* may play a role in food-cobalamin malabsorption.

The biological plausibility of this hypothesis is further supported by data on negative effects on serum cobalamin levels by long-term treatment with proton pump inhibitors [13,61]; in the most recent study performed in Kosovo, after 12 months of proton pump inhibitors treatment, low vitamin B_12_ serum levels were found in about 2.9% of subjects [62]. A two-year or longer treatment with proton pump inhibitors or the formerly used antisecretory drug, H_2_-receptor blocker, was associated with an increased risk for vitamin B_12_ deficiency (OR, 1.65; 95% CI, 1.58–1.73 and OR, 1.25; 95% CI, 1.17–1.34, respectively) in a dose-dependent fashion [63].

However, in the above-cited systematic review [4], the eventual mechanisms linking micronutrient deficiency, *Helicobacter pylori* infection, and low gastric acid secretion were addressed only in part of the included studies. Therefore, the role of reduced gastric acid secretion possibly leading to low cobalamin levels in the presence of *Helicobacter pylori* infection is still poorly investigated, and mechanisms different from impaired gastric acid secretion linking together *Helicobacter pylori* and low cobalamin levels may not be excluded.

In contrast to the possible functional inhibition of gastric acid in corpus-involving *Helicobacter pylori* gastritis, atrophic gastritis involving the corpus mucosa due to gastric autoimmunity or, again, to long-standing *Helicobacter pylori* infection may lead to irreversible damage of gastric oxyntic mucosa leading to hypochlorhydria and concomitant impaired intrinsic factor secretion. This, over time, may result initially in subclinical vitamin B_12_ deficiency without anemia, which often generally takes about 10–12 years, to clinically manifest pernicious anemia [12,64]. From a clinical point of view, besides anemia, vitamin B_12_ deficiency may also lead to neurological complications ranging from paraesthesia, numbness, abnormal proprioception, ataxia, cognitive impairment, to mood disorders and frank psychosis. Unfortunately, these neurological alterations, when not promptly recognized and treated, may not be reversible after cobalamin supplementation [54,65].

*Helicobacter pylori*-related corpus atrophic gastritis, also after successful cure of *Helicobacter pylori* infection, is infrequently reversible [66,67]. Most commonly, in patients with corpus atrophic gastritis and pernicious anemia, lifetime oral or more often parenteral vitamin B_12_ supplementation is necessary. Figure 1 shows the main causes of vitamin B12 deficiencies.

## 5. The Role of the Stomach in Ascorbate Deficiency

Ascorbic acid, the reduced form of vitamin C, is a water-soluble antioxidant compound, important for scavenging nitrite-derived mutagens and conferring protection against carcinogenesis [68]. Diet is the most important determinant of plasma ascorbic acid levels in humans as they are not able to synthesize ascorbic acid because of the loss of gulonolactone oxidase due to several mutations during evolution, an enzyme necessary for the last step of vitamin C biosynthesis [69]. After absorption, vitamin C is actively secreted and concentrated in the gastric juice in its reduced form of ascorbic acid resulting in higher levels in the gastric juice than in the plasma [70,71]. Ascorbic acid is also an important iron absorption promoter, and its reduced bioavailability may, in turn, have a negative rebound on iron absorption. Intriguingly, it has been proposed that similar to the intrinsic factor necessary for vitamin B_12_ absorption, the “intrinsic factor” ascorbic acid is needed for iron absorption [72]: ascorbic acid converts the ferric iron to its ferrous form that remains soluble in the alkaline duodenal environment and forms chelates with ferric chloride that is stable at a pH higher than 3 [73].

Based on serology, in *Helicobacter pylori*-positive compared to -negative subjects, lower plasma ascorbic acid levels have been shown, even after correction for smoking and diet, factors able to importantly influence ascorbic acid levels [74]. Diet, in particular an adequate intake of fresh fruit and vegetables, could account for different ascorbic acid levels in different countries. Several studies have related *Helicobacter pylori* infection to decreased vitamin C levels, both in gastric juice and plasma [75,76,77,78]. A systematic review on the possible link between *Helicobacter pylori* infection and reduced micronutrients beyond iron showed a positive association between *Helicobacter pylori* positivity and low ascorbic acid levels in the plasma (SMD 0.2) and in the gastric juice (SMD 1.1), and recovery after the cure of infection in the gastric juice (SMD 1.4) [4], but not in plasma. This suggests that the cure of *Helicobacter pylori* infection may recover the normal transport or secretion of ascorbic acid from plasma into gastric juice [79].

The mechanisms by which *Helicobacter pylori* infection may impair the homeostasis of ascorbate and, in particular, the decrease of plasma ascorbic acid concentration, has not been clarified so far, but the following mechanisms have been proposed: lower bioavailability, insufficient vitamin C intake, hypochlorhydria, higher consumption due to increased active secretion from plasma to gastric juice in the attempt to restore the positive juice/plasma ratio, and *Helicobacter pylori*-associated oxidants accelerating ascorbic acid degradation [79]. Reduced systemic bioavailability of ascorbic acid was shown in subjects infected with *Helicobacter pylori* and was independent of diet [74]. A role of drug-induced reduced gastric acid secretion was proved by a study reporting that a proton pump inhibitor given for 4 weeks reduced plasma vitamin C levels in *Helicobacter pylori*-infected and non-infected subjects with similar dietary intake suggesting a reduced bioavailability of dietary vitamin C in these conditions [80]. After the cure of *Helicobacter pylori,* infection reduced plasma vitamin C levels have been reported suggesting an enhanced active transport of ascorbic acid to regain the high ratio of gastric juice to plasma ascorbic acid [75]. The inflammatory process related to *Helicobacter pylori* infection with high production of radical oxygen species may lead to increased ascorbate consumption [79,81].

Thus, the observed positive association between *Helicobacter pylori* infection and ascorbic acid, but also cobalamin and iron levels, may imply common mechanisms involving hypochlorhydria due to impaired gastric acid secretion as a result of *Helicobacter pylori*-related inflammatory damage involving corpus oxyntic mucosa [5]. A negative correlation has been reported between ascorbic acid concentration in the gastric juice and the acute inflammatory score in the antral mucosa [82]. While no differences between the ascorbic acid concentration in the gastric juice between subjects with antral-confined gastritis and healthy subjects (*Helicobacter pylori*-negative) were reported, in contrast, in subjects with corpus-involving *Helicobacter pylori* gastritis, the ascorbic acid levels were higher and intragastric pH was increased [77]. This may imply that ascorbic acid, a very unstable compound in a non-acidic environment, is degraded to the less active dehydro-ascorbic acid, and this may occur when the acid-secreting oxyntic corpus mucosa is damaged by an inflammatory process to *Helicobacter pylori* infection eventually leading to atrophy with consequent impaired gastric acid secretion [5].

In autoimmune atrophic gastritis, the prevalence of deficiency of ascorbic acid is not known [64]. The metabolic relationship between ascorbic acid and vitamin B_12_ has been investigated showing lower plasma ascorbate levels before cobalamin supplementation that improved after vitamin B_12_ treatment [83]. Further studies are needed to define the occurrence and the eventual clinical implications of ascorbate deficiency in individuals with autoimmune atrophic gastritis.

## 6. Calcium, Reduced Bone Mineral Density, Fractures, and Impaired Gastric Acid Secretion

Calcium absorption begins thanks to the intragastric acid pH favoring the dissolution of calcium salts to form soluble calcium chloride, a step facilitating proper absorption of this ion in the proximal small intestine [84,85]. The crucial role of gastric HCl in calcium absorption was highlighted by Recker R. reporting that its absorption, measured by a modified double-isotope procedure, was lower in patients with achlorhydria compared to healthy individuals [86]. In another study, measuring calcium absorption in postmenopausal women with pernicious anemia and normal postmenopausal women, no significant differences were found, suggesting that the issue of calcium malabsorption in patients with autoimmune atrophic gastritis remains controversial [87]. However, the same authors also reported a significant loss of bone mineral density in postmenopausal women affected by pernicious anemia, speculating that the loss of cancellous bone must be caused by some mechanisms yet to be identified [87]. However, the prevalence of osteopenia and osteoporosis in autoimmune atrophic gastritis is currently unknown.

In the context of calcium absorption, the role of vitamin D cannot be excluded [88]. Antico et al. reported significantly lower 25-OH vitamin D levels in patients with autoimmune atrophic gastritis than in non-autoimmune gastritis atrophic gastritis, or in the general population, supporting the hypothesis that hypovitaminosis D might be a risk factor for the development of autoimmune diseases [89]. More recently, Massironi et al. reported an increased prevalence of hyperparathyroidism secondary to vitamin D deficiency in autoimmune atrophic gastritis patients [90], even if the post postmenopausal status is unknown.

Regarding calcium absorption and the impact of gastric hypochlorhydria on its homeostasis, the possible role of anti-secretory drugs needs to be considered; these pharmacological agents, by reducing gastric acid secretion, likely affect calcium absorption and increase the risk of osteoporosis and bone fractures [85]. Indeed, a previous study suggested that omeprazole might impair osteoclast activity, which is dependent on H+/K+-ATPase resembling the proton pump placed on parietal cells [91].

Several authors showed the presence of a positive association between the use of proton pump inhibitors (PPIs) and hip fractures in postmenopausal women [92], men [93], elderly subjects [94,95], and also in patients with type 2 diabetes [96]. Recently, it has been observed that PPI use was associated with bone fractures in a dose-dependent fashion and also in patients with liver cirrhosis [97].

A systematic review and meta-analysis by Zhou B et al. aimed to identify the relationship between PPIs and the risk of fracture and found a moderately increased risk for hip fracture (RR = 1.26; 95% CI: 1.16–1.36) but with significant heterogeneity (*p* < 0.001; I^2^ = 71.9%) for spine fracture (RR = 1.58; 95% CI: 1.38–1.82) and any-site fracture (RR = 1.33; 95% CI: 1.15–1.54), without significant difference between short-term (<1 year) and long-term use (>1 year) [98]. This positive association between PPIs and risk of hip fracture was also confirmed by a more recent meta-analysis of observational studies [99].

These data taken together allow the conclusion that the chronic use of PPIs is linked to increased risk of bone fractures likely by impairing calcium homeostasis, at least in particular subsets of patients like postmenopausal women, elderly and fragile subjects. Thus, inappropriate PPI prescriptions should be accurately avoided, especially in patients who are at increased risk of bone fractures.

## 7. Other Micronutrient Deficiencies Related to Drug-Induced Hypochlorhydria

### 7.1. Magnesium Deficiency and Anti-Secretory Drugs

Anti-secretory drugs, particularly PPIs are a mainstay in the treatment of acid-related disorders, such as gastroesophageal reflux disease and dyspeptic symptoms. Even if these drugs are generally considered safe, in the last years, several studies reported adverse effects associated with long-term use of PPIs [100].

It has been observed that PPI use may favor hypomagnesemia with the onset of muscle cramps, paraesthesia, and eventually the potentially serious cardiac arrhythmias. This association was firstly reported in two patients in 2006 [101], followed by the publication of other case reports and case series [102,103,104,105,106], and a recent cross-sectional study [107]. However, available data are conflicting with other authors not confirming this association. Chowdhry et al. reported no significant differences in mean magnesium levels between PPI users and PPI non-users (*p* = 0.40) regardless of PPI dosage or concomitant diuretics assumption [108].

A systematic review and meta-analysis by Park CH et al. summarized the evidence regarding the association between the use of PPIs and the risk of developing hypomagnesemia [109]. Authors included 115,455 patients, reporting that the median proportion of patients with hypomagnesemia was 27.1% (range: 11.3–55.2%) and 18.4% (range: 4.3–52.7%) in patients taking and not taking PPIs respectively. On meta-analysis, pooled OR for PPIs use was found to be 1.78 (95% CI 1.08–2.92), but the significant heterogeneity among studies (I^2^ = 98%) do not allow authors to draw definitive conclusions [109]. Similar conclusions were drawn in a more recent systematic review and meta-analysis indicating that the use of PPIs increased the risk of hypomagnesemia (RR, 1.44, 95% CI, 1.13–1.76), again with high heterogeneity (I^2^, 85.2%); subgroup analysis revealed that the use of PPIs was neither associated with the incidence of hypomagnesemia in outpatients (RR, 1.49; 95% CI, 0.83–2.14; I^2^ 41.4%) nor hospitalized patients (RR, 1.05; 95% CI, 0.81–1.29; I^2^ 62.1%), respectively [110]. However, the highly variable follow-up period of the studies impairs the evidence of the possible role of PPIs in inducing hypomagnesemia and related clinical symptoms.

The mechanism underlying the hypothesized association between PPI use and hypomagnesemia is not yet elucidated. It has been speculated that PPIs-induced hypochlorhydria may reduce the active transport of magnesium, affecting the affinity of the transient receptor potential (TRP) ion channels TRPM6 and TRPM7 for magnesium [111,112]. Further controlled studies are needed to better understand this supposed association.

In the meantime, it seems advisable and cautious to regularly assess serum magnesium levels in patients of long-term antisecretory treatment which cannot be withdrawn to monitor for the eventual onset of magnesium deficiency and to carefully pay attention to specific symptoms or signs of hypomagnesemia in these patients.

### 7.2. Other Micronutrients

The chronic use of antisecretory agents, PPIs, and H_2_ blockers have also been implicated in the deficiency of other micronutrients such as iron and vitamin B_12_. The link between long-term antisecretory drugs and bone fractures and/or calcium deficiency is separately discussed (see above). The supposed mechanisms for iron and vitamin B_12_ malabsorption are similar to those discussed for corpus atrophic gastritis and *Helicobacter pylori*-related superficial gastritis involving the corpus mucosa and are related to reduced gastric acid and increased intragastric pH impairing the food-bound absorption of cobalamin and the dissociation, solubilization, and reduction of iron salts ultimately hindering the formation of the complex iron-ascorbate needed for absorption [5,13].

Several reports showed data that long-term use of antisecretory drugs can lead to vitamin B_12_ and iron deficiency [13], but provided data are still conflicting [113,114,115,116].

Concerning vitamin B_12_ deficiency, several studies reported a positive association between long-term use of antisecretory drugs, in particular PPIs, and reduced serum levels of cobalamin, and a higher prevalence of vitamin B_12_ deficiency, especially in elderly subjects, but to date, this is not definitively established. Further studies are needed to draw definite conclusions. With the currently available data, serum vitamin B_12_ levels should probably be monitored in subsets of long-term users of PPIs, in the elderly, and subjects with Zollinger-Ellison syndrome or other gastric hypersecretory states [13].

In considering iron absorption, some reports on patients with long-term use of PPIs found a positive association between reduced iron absorption and drug-induced hypochlorhydria, while others did not confirm these results [117,118,119,120]. In particular, in a study on patients with Zollinger Ellison syndrome who were on PPI therapy for at least six years, no iron deficiency was documented but decreased vitamin B_12_ levels were observed [120].

As mentioned above, it has been reported that omeprazole administered for 4 weeks lowered plasma vitamin C levels in *Helicobacter pylori*-infected and non-infected subjects with similar amounts of vitamin C in the diet. Thus, drug-induced hypochlorhydria may play a role in reducing the bioavailability of dietary vitamin C [80].

What is lacking up to date is an ascertained relationship between the long-term use of antisecretory drugs and serious clinical implications for these patients and precise indications of which subsets of subjects need to be monitored over time to prevent micronutrient deficiencies and related complications.

## 8. Gastric Surgery and Micronutrient Deficiencies

Virtually all gastric surgical procedures may over time lead to micronutrient deficiencies as a result of the above-exposed link between a healthy stomach and the homeostasis of several micronutrients such as iron, cobalamin, ascorbic acid, and magnesium amongst the most important.

In the past years, before the discovery of *Helicobacter pylori* as the main etiological agent of peptic ulcer, gastric surgery was frequently performed due to otherwise untreatable or life-threatening recurrent or bleeding duodenal peptic ulcers and less frequently gastric ulcers. In the present, total or partial gastric resection according to Billroth I or II is performed mainly for surgical treatment of gastric neoplasia (gastric adenocarcinoma, type III gastric neuroendocrine tumors, or less common histophenotypes of gastric neoplastic lesions) or complicated, non-endoscopically treatable peptic ulcers. In these last years, surgical procedures involving the stomach are increasingly performed to obtain voluntary weight loss (bariatric surgery) and the most commonly proposed surgical options are sleeve gastrectomy, Roux-en-Y gastric bypass, or biliopancreatic diversion [121,122]. Figure 2 represents a graph of the most common gastric surgical procedures.

In the case of total gastrectomy, the complete absence of the gastric mucosa and thus gastric acid and intrinsic factor secretion leads over time to impaired iron and vitamin B_12_ absorption, once body stores are depleted. Addtionally, in the case of partial gastrectomy due to the loss of the gastrin-producing antral mucosa (Billroth I, II, Roux-en-Y gastric bypass, biliopancreatic diversion) or the loss of part of the oxyntic mucosa (sleeve gastrectomy), the regular absorption of the hematopoietic micronutrients iron and vitamin B_12_ is impaired [123]. In these patients, when not regularly monitored with blood tests, chronic anemia may develop. A retrospective study on long-term gastric cancer survivors without recurrence or metastasis after gastrectomy showed a cumulative incidence rate of anemia after surgery linearly increasing from 18.7% in the first year to 39.5% in the fifth year [124]. Another study explored the cumulative incidence of anemia after gastrectomy in patients with early-stage gastric cancer with a long life expectancy and reported that among 566 patients who did not have anemia preoperatively, 240 (42.4%) experienced anemia at least once during the 5 years after gastrectomy [122].

In the case of patients with a peptic ulcer who underwent partial gastrectomy or in subjects who underwent sleeve gastrectomy in whom an eventually persistent *Helicobacter pylori* infection at the remnant gastric corpus mucosa has not been detected and cured, another additional mechanism of micronutrient malabsorption may be the inflammatory injury of the persistent infection which may aggravate the entity of malabsorption and possibly lead over time to atrophy of the remnant oxyntic mucosa.

Concerning patients with gastric cancer, the assessment of eventual micronutrient deficiencies is also important prior to gastric surgery, as the presence of the neoplasia itself may lead to impaired homeostasis of some micronutrients due to different mechanisms. Iron deficiency with or without anemia may occur in gastric cancer patients due to bleeding of the neoplastic lesion, most often in a context of misdiagnosed corpus atrophic gastritis with concomitant iron and/or cobalamin malabsorption [125]. The presence of gastric neoplasia may also lead to difficulty in nutrition due to pain after food ingestion and/or food refusal due to loss of appetite as a consequence of the oncological disease resulting, over time, in a reduced intake of macro- and micronutrients [125]. A retrospective Canadian study on 126 gastric cancer patients showed that 40% had iron deficiency anemia at the time of gastric cancer diagnosis [126]. Another study reported that in patients with gastric cancer, preoperative anemia was significantly associated with more perioperative transfusions, postoperative complications and worse prognostic nutritional index, preoperative weight loss and performance status, and was an independent prognostic factor in TNM stage III gastric cancer patients affecting overall survival [127], albeit in this study the type of anemia was not specified.

Many experimental and clinical studies have reported that vitamin D may be linked to gastric cancer, albeit the mechanisms are unclear [128]. Hypovitaminosis D has been related also to autoimmune atrophic gastritis, a precursor condition of gastric cancer [89,90]. For example, vitamin D3 was reported to induce apoptosis in gastric cancer cells and suggested to regulate the Hedgehog signaling pathway in gastric cancer cells [129,130]. In patients with gastric cancer, an inverse relationship between gastric cancer stage and lymph node metastasis and pretreatment vitamin D levels were reported: higher serum levels of 25-hydroxyvitamin D (>50 nmol/L) were observed to be linked to a higher overall survival rate than lower levels (<50 nmol/L, *p* = 0.018). Further, serum levels of 25-hydroxyvitamin D were an independent prognostic factor of gastric cancer (*p* = 0.019) [131]. Anyhow, a systematic review failed to confirm a relationship between vitamin D and gastric cancer [132]. But recently, a Korean study assessed the association between vitamin D concentrations and gastric cancer prevalence reporting data on serum 25-hydroxyvitamin D concentrations as a biomarker of vitamin D status of 33,119 adults [133]. They showed in an adjusted model an OR of 0.84 (95% CI: 0.72, 0.98) for gastric cancer with a 5-ng/mL increment in total vitamin D suggesting that higher vitamin D concentrations are associated with a lower risk of gastric cancer in Korean adults. Thus, while awaiting further studies clarifying this issue, it seems advisable to preoperatively assess the micronutrient status in patients with gastric cancer, in particular iron and vitamin D, as the impaired homeostasis might affect the outcome of the patient.

Micronutrient deficiencies after bariatric surgery [134], and in particular sleeve gastrectomy, are a common phenomenon [121,135]. Notwithstanding 64% of the included patients had a pre-operative correction of micronutrient deficiencies, 12 months after surgery without any supplementation, 29%, 32% 63%, 18%, and 5.4% had iron, cobalamin, vitamin D, calcium, and magnesium deficiency, respectively [136]. A meta-analysis on patients who underwent bariatric surgery (irrespective of the surgical technique, sleeve gastrectomy or Roux-en-Y gastric bypass) showed that patients without prophylactic iron or vitamin B_12_ supplementation had a 1.8- (95% CI 0.7 to 4.2) and 6.3-fold (95% CI 2.7 to 14.6) higher probability of becoming deficient after surgery when compared to those who were supplemented (follow-up 12–60 months) [137]. Another meta-analysis on patients with Roux-en-Y gastric bypass surgery reported similar results showing that after surgery the proportion of patients with anemia increased from 12.2% at baseline to 20.9% and 25.9% at 12 and 24 months follow-up, the proportion of patients with ferritin deficiency increased from 7.9% at baseline to 13.4% and 23.0% at 12 and 24 months, and those with vitamin B_12_ deficiency increased from 2.3% at baseline to 6.5% at 12 months, while folate deficiency seemed not to occur [138]. Other micronutrient deficiencies are less investigated and documented.

Thus, subjects who underwent gastric surgery should be regularly monitored to timely detect the onset of micronutrient deficiencies that need to be supplemented to avoid potentially serious complications as anemia or neuropathy.

## 9. Possible Role of Gastric Dysbiosis

Amongst others, gastric acidity also has the primary function of a bactericidal defensive barrier. The progressive destruction of acid-secreting parietal cells is characteristic of corpus atrophic gastritis leading to a gradual hypochlorhydric state. The consequent increase of pH leads to the weakening of the gastric acid defensive barrier resulting in alterations of the gastric microbiota composition with the potential overgrowth of other bacterial strains than *Helicobacter pylori* [17]. Doubtlessly, *Helicobacter pylori* still play a prominent role as the main responsibility for gastric disease and long-term complications. However, due to the peculiar intra-gastric environment characterized by impaired gastric acid secretion, the gastric microbiota composition in patients with autoimmune and non-autoimmune corpus atrophic gastritis has been reported to be different from subjects with a normal acidic healthy stomach allowing the survival and colonization of other bacterial strains than *Helicobacter pylori* [139,140], and gastric dysbiosis is supposed to be related to the increased risk of gastric cancer in this condition [17,141].

Many microorganisms may compete for essential minerals and metals with the host, and there is a body of evidence showing that *Helicobacter pylori* is genetically equipped to sequester and facilitate the acquisition of iron and other essential metals. For example, the control of iron homeostasis is mediated by the ferric uptake regulator, which essentially regulates the transcription of genes involved in iron acquisition and storage in response to changes in iron availability [142]. Thus, it can be supposed that, in the presence of gastric dysbiosis, unusual bacteria colonizing the hypochlorhydric stomach may be able to interfere and compete with the homeostasis of essential minerals and metals, but to date studies on this intriguing topic are lacking and a specific role of gastric dysbiosis in the impairment of micronutrient bioavailability or absorption has not been reported so far.

## 10. Conclusions

Deficiencies of several micronutrients as iron, cobalamin, calcium, magnesium, and ascorbic acid may lead to potentially serious consequences when not promptly diagnosed and treated. This is particularly true for the hematopoietic micronutrients iron and cobalamin whose deficiency may lead to anemia with slow onset over time, which may be potentially dangerous in elderly subjects with cardiac comorbidities.

A key role in the homeostasis of these micronutrients is played by the gastric acid secretion and all pathological and iatrogenic conditions leading to impairment of gastric acid secretion may virtually lead to the deficiency of these micronutrients. First of all, for the high prevalence, *Helicobacter pylori*-related gastritis involving the corpus mucosa, followed by the presence of autoimmune or *Helicobacter pylori*-related corpus atrophic gastritis should be kept in mind. During the diagnostic work-up for micronutrient deficiencies, particular attention should be paid to the long-term use of anti-secretory drugs, in particular proton-pump inhibitors, potentially leading over time to micronutrient malabsorption due to prolonged inhibition of gastric acid secretion. Last, but not least, previous gastric surgery also needs to be taken into consideration, formerly performed frequently for peptic ulcers and still performed for gastric cancer, but in recent years, performed increasingly for bariatric surgical procedures for weight loss in morbidly obese patients such as sleeve gastrectomy, Roux-en-Y gastric bypass, or biliopancreatic diversion with duodenal switch. All these surgical procedures may negatively impact gastric acid secretion and lead over time to micronutrient deficiency, in particular iron and/or cobalamin deficiency with a consequent risk of anemia. In the case of a previous peptic ulcer or gastric cancer, a subsequently persistent *Helicobacter pylori* infection may play a concomitant role.

As shown in Table 2, in patients with micronutrient deficiency, we recommend excluding first of all *Helicobacter pylori* infection and autoimmune or *Helicobacter pylori*-related corpus atrophic gastritis, also in subjects who previously underwent gastric surgery (when this has not been done before surgery). This can be performed most efficaciously by gastroscopy with multiple biopsies taken in the antral (when present) and corpus mucosa sent for histopathological assessment. When corpus atrophic gastritis is diagnosed, patients should be advised to enrich their intake of vitamin C by eating at least twice daily fresh fruits and vegetables, and when cobalamin or iron deficiency is detected, life-long supplementation is required. In subjects who have undergone previous gastric surgery, biochemical monitoring of early onset of micronutrient deficiency is suggested which, once diagnosed, needs to be adequately treated by oral or eventually parenteral supplementation.

In patients on long-term treatment with proton pump inhibitors, the correct indication should be checked, and when these drugs cannot be withdrawn, biochemical monitoring of iron, cobalamin, calcium, and magnesium deficiencies is advisable.

In conclusion, patients with isolated or more particularly, several concomitant micronutrient deficiencies such as iron, cobalamin, calcium, and/or magnesium, amongst other possible diagnoses, should always keep in mind the stomach!

## Figures and Tables

**Figure 1 nutrients-13-00208-f001:**
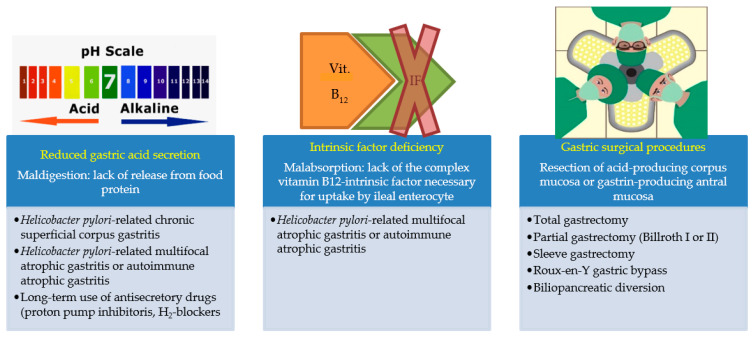
Vitamin B_12_ deficiency may be caused, besides reduced intake, by three different conditions.

**Figure 2 nutrients-13-00208-f002:**
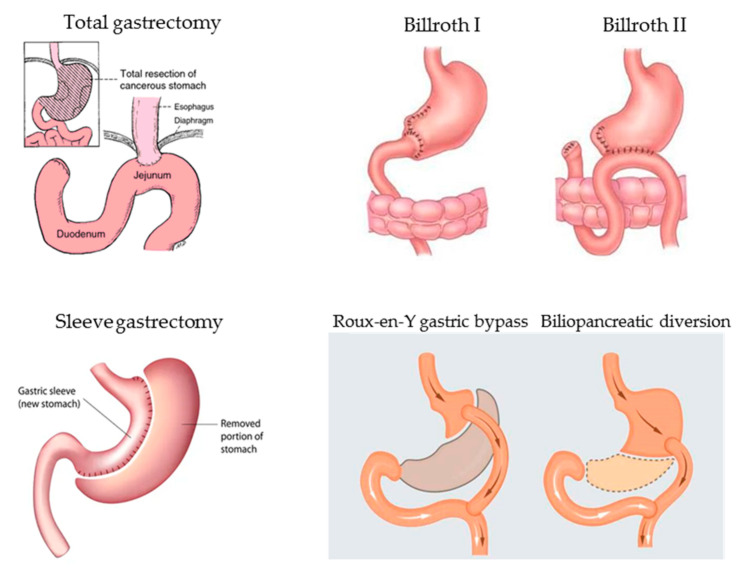
Main gastric surgical procedures potentially leading over time to micronutrient deficiencies due to resection or exclusion of functionally active gastric mucosa necessary for gastric acid and intrinsic factor secretion, in turn, essential for correct micronutrient absorption.

**Table 1 nutrients-13-00208-t001:** Principals stimulants and inhibitors of human gastric acid secretion and their mechanisms of action.

	Mechanisms of Action
**Stimulants of gastric acid secretion**	
Histamine	Histamine is released by ECL cells. It stimulates HCl secretion directly (binding H2 receptors, placed on parietal cell, coupled with activation of adenylate cyclase and generation of 3′5′-cyclic adenylate cyclase), and indirectly (inhibiting somatostatin release from D cells).
Gastrin	Gastrin is released by antral G cells. It stimulates HCl secretion directly (acting on the CCK2 receptor and activating the release of intracellular calcium) and indirectly (stimulating histamine release)
Acetylcholine (ACh)	ACh is released by postganglionic enteric neurons. It acts directly (binding M3 receptors, placed on parietal cell, coupled with increasing intracellular calcium) and indirectly (inhibiting somatostatin secretion).
**Inhibitor of gastric acid secretion**	
Somatostatin	Somatostatin is released by D cells. It acts directly on parietal cells and indirectly by inhibiting histamine release.

ECL = enterochromaffin-like.

**Table 2 nutrients-13-00208-t002:** Suggestions of a To-Do List to check patients with micronutrient deficiencies such as iron, cobalamin, calcium, magnesium, and/or ascorbic acid.

To-Do List
To look for and to eventually treat *Helicobacter pylori*
To consider the presence of corpus atrophic gastritis and eventually perform gastroscopy with antral and corpus biopsies To advise vitamin C intake from fresh fruits and vegetables in patients with corpus atrophic gastritis
To check for previous gastric surgery
To routinely monitor patients with any previous gastric surgery for micronutrient deficiency
To reassess the indication of long-term of proton pump inhibitors

## Data Availability

Data sharing not applicable.

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
