# Peer review of "Common Pitfalls in the Management of Patients with Micronutrient Deficiency: Keep in Mind the Stomach"

_nutrients, 2021, doi:10.3390/nu13010208_

Round 1

Reviewer 1 Report

It is a valuable paper that summarizes what is currently known for patients with micronutrient deficiency by focusing on the stomach and draws attention to medical personnel.

I haven't pointed out any major corrections, but please check the following points to improve the appearance of the text.

  1. Regarding line 124, I think that "human" means an infectious disease in humans, but please confirm that point.
  2. For 226 lines, the subtitles are duplicated, so I think it's a good idea to delete them.
  3. For line 334, the "and" at the end of the sentence doesn't make sense, so I think it's a good idea to delete it.
  4. For line 369, I think "6.1." Is "6.2."
  5. I think it's a good idea to add a period at the end of lines 386, 386, and 391.

Author Response

Reviewer 1

Comments and Suggestions for Authors

It is a valuable paper that summarizes what is currently known for patients with micronutrient deficiency by focusing on the stomach and draws attention to medical personnel.

I haven't pointed out any major corrections, but please check the following points to improve the appearance of the text.

1.Regarding line 124, I think that "human" means an infectious disease in humans, but please confirm that point.

Response: Thank you for the opportunity to rephrase the sentence. We modified the sentence accordingly to your suggestion (see line 128). 

2.For 226 lines, the subtitles are duplicated, so I think it's a good idea to delete them.

Response: Accordingly with your suggestion we deleted the subtitles (see line 230).   

3.For line 334, the "and" at the end of the sentence doesn't make sense, so I think it's a good idea to delete it.

Response: Accordingly with your suggestion we deleted the “and” at the end of the sentence (see line 338).

4.For line 369, I think "6.1." Is "6.2."

Response: Thank you for your suggestion. We noted that in the previous version of the manuscript an error occurred in the paragraph’s numeration (paragraph number 4 was doubled, see line 174 and 240). We have numbered paragraphs again, also keeping in mind your comment (lines 339, 373).

5.I think it's a good idea to add a period at the end of lines 386, 386, and 391.

Response: Thank you for you comment. We added a period at the end of lines 390, 393 and 395.

Reviewer 2 Report

A comprehensive review of gastric acid secretion and the role of the stomach in iron, cobalamin, and magnesium homeostasis. 

Author Response

Reviewer 2

Comments and Suggestions for Authors

A comprehensive review of gastric acid secretion and the role of the stomach in iron, cobalamin, and magnesium homeostasis. 

Response: We are very pleased for your comment and we kindly thank you for appreciating our work.

Reviewer 3 Report

In the manuscript entitled "Common pitfalls in the management of patients with micronutrient deficiency: keep in mind the stomach" by Carabotti quite elegantly discuss the roles of stomach in the homeostasis of several essential micronutrients. 

Major concern:

  1. In Chapter 7 entitled 'Gastric surgery in micronutrient deficiencies', although the authors discuss the context of gastric cancer in micronutrient deficiencies, it is important to mention the role of gastric cancer independent of the consequent (if any) surgical interventions. There is a significant number of literatures available that mention the involvement of the gastric cancer in altered micronutrient homeostasis.
  2. Recently there is a surge of novel studies on gut microbiota (including gastric ones) emphasizing their various roles on systemic and local pathophysiology. Discussion of these factors in light of newer findings beyond H. pylori would be very useful for the readership. 

Minor concern:

    1. For the information mentioned in Table 1, please indicate appropriate citations, although they are very well known textbook references. 

    2. In Chapter 3, page 4, lines 113-4, reference#28: please elaborate the information with a more recent citation as we have significantly updated information over the last two decades. 

Author Response

Reviewer 3

In the manuscript entitled "Common pitfalls in the management of patients with micronutrient deficiency: keep in mind the stomach" by Carabotti quite elegantly discuss the roles of stomach in the homeostasis of several essential micronutrients. 

Major concern:

1.In Chapter 7 entitled 'Gastric surgery in micronutrient deficiencies', although the authors discuss the context of gastric cancer in micronutrient deficiencies, it is important to mention the role of gastric cancer independent of the consequent (if any) surgical interventions. There is a significant number of literatures available that mention the involvement of the gastric cancer in altered micronutrient homeostasis.

Response: We thank the Reviewer for his interesting suggestion certainly improving the paper. Accordingly, we added a subparagraph on the end of para 8 (see lines 443-475).

2.Recently there is a surge of novel studies on gut microbiota (including gastric ones) emphasizing their various roles on systemic and local pathophysiology. Discussion of these factors in light of newer findings beyond H. pylori would be very useful for the readership. 

Response: Thanks again for this suggestion with which we fully agree. We added a new sentence in the introduction section (see lines 57-61) and a new paragraph on this topic (see lines 494-516). We think that this additional information completes the review improving it.

Minor concern:

1.For the information mentioned in Table 1, please indicate appropriate citations, although they are very well known textbook references. 

Response: Accordingly with your suggestion, we added citations for the Table 1 (see line 99), adding also a new reference [num. 28: Schubert M. L. (2017). Physiologic, pathophysiologic, and pharmacologic regulation of gastric acid secretion. Current opinion in gastroenterology33(6), 430–438. https://doi.org/10.1097/MOG.0000000000000392)]

2.In Chapter 3, page 4, lines 113-4, reference#28: please elaborate the information with a more recent citation as we have significantly updated information over the last two decades. 

Response: Accordingly with your suggestion, we added a new sentence and a new reference (see lines 118-120).

Round 2

Reviewer 3 Report

The authors have convincingly addressed and included the relevant concerns put by this reviewer. The manuscript has significantly improved. Many thanks!